# Resveratrol Ameliorates Chronic Stress in Kennel Dogs and Mice by Regulating Gut Microbiome and Metabolome Related to Tryptophan Metabolism

**DOI:** 10.3390/antiox14020195

**Published:** 2025-02-09

**Authors:** Zhaowei Bian, Ziyang Li, Hao Chang, Jun Luo, Shiyan Jian, Jie Zhang, Peixin Lin, Baichuan Deng, Jinping Deng, Lingna Zhang

**Affiliations:** Laboratory of Companion Animal Science, Department of Animal Science, South China Agricultural University, Guangzhou 510642, China; bzw02052021@163.com (Z.B.); 13966379603@163.com (Z.L.); 19584810169@163.com (H.C.); 16676783813@163.com (J.L.); janshiyan1029@163.com (S.J.); zjie1103@stu.scau.edu.cn (J.Z.); lpp816816@163.com (P.L.); dengbaichuan@scau.edu.cn (B.D.)

**Keywords:** dog, chronic stress, gut microbiota, tryptophan, metabolomics

## Abstract

Chronic stress poses threats to the physical and psychological well-being of dogs. Resveratrol (Res) is a polyphenol with antidepressant properties and has rarely been studied in dogs. This study aimed to investigate the stress-relieving effects and underlying mechanism of Res in dogs. Dogs were fed a basal diet supplemented with Res for 35 days. The fecal microbiota of the dogs was cultured with Res in vitro. The results show that Res improved the stress-related behaviors and increased the serum levels of 5-hydroxytryptamine (5-HT), brain-derived neurotrophic factor (BDNF), immunoglobulin A, and antioxidant capacity in dogs. Res downregulated the hormones of the hypothalamic–pituitary–adrenal axis. The abundance of butyric-producing bacteria, like *Blautia*, increased, while the growth of *Fusobacterium* related to gut inflammation was inhibited in the Res group. A higher content of fecal butyric acid was observed in the Res group. The metabolome indicated that Res increased the fecal and serum levels of tryptophan (Trp) and decreased the consumption of Trp by microorganisms. A chronic unpredictable mild stress mouse model was established, and Res was administered for 35 days. The results show that Res ameliorated the stress-related behavior and increased the levels of Trp and 5-HT in the whole brains of mice. The relative mRNA expression of genes associated with the tight junction protein, aryl hydrocarbon receptor, and Trp transporters in the colon were upregulated. In conclusion, Res could ameliorate canine stress by increasing 5-HT, BDNF, and the antioxidant capacity and improving the immune function and stress response, which was attributed to the role of Res in the restructuring of gut microbiota and the modulation of tryptophan metabolism.

## 1. Introduction

Captive dogs in a kennel environment can easily suffer from chronic stress as a result of factors such as a restricted living space, environmental noise, inconsistent caretaking routine, and a lack of social interaction [1]. Long-term exposure to stressful conditions might trigger anxiety in dogs, which increases the risk of gastrointestinal and urological disease. Behavioral training and the administration of psychotropic drugs (e.g., fluoxetine) and appeasing pheromones have been introduced to relieve stress in dogs. But the side effects of psychotropic drugs and the lack of feasibility and consistency in the therapeutic effect of behavior correction and pheromone treatments have limited their widespread applications [2,3].

Resveratrol (Res) is a polyphenol with potent anti-inflammatory, antioxidant, and antidepressant properties. A previous study found that Res could alleviate rat depression-like behaviors through the elevation of brain-derived neurotrophic factor (BDNF) [4]. Gu et al. (2019) proposed that the mechanism of the function of Res in relieving stress was similar to the fluoxetine, which could increase 5-hydroxytryptamine (5-HT) in the prefrontal cortex [5]. Hormones along the hypothalamic–pituitary–adrenal (HPA) axis, including corticotropin-releasing hormone (CRH); adreno-cortico-tropic-hormone (ACTH); and a glucocorticoid (GC), mainly cortisol, are key components of the stress response. Res could alleviate stress and psychiatric disorders by downregulating the gene expression of CRH and blood concentration of corticosterone [6]. Plenty of evidence manifested showing that tryptophan (Trp) dysmetabolism was associated with depression and neurodegenerative diseases [7]. In addition, recent studies on the microbiota–gut–brain (MGB) axis suggested that gut microbes could participate in the development of anxiety and depression through the regulation of short-chain fatty acids (SCFAs) and Trp metabolism [8]. The levels of 5-HT in the central nervous system (CNS) are closely related to the stress state of the body, and 5-HT is one of the metabolic products of tryptophan in the brain [7]. The tryptophan in the CNS originates from the absorption of tryptophan from food by the gut. Evidence shows that gut microbiota can regulate the availability of amino acids in the gut, thereby affecting the levels of central amino-acid-derived neurotransmitters (5-HT, dopamine) in an amino-acid-dependent manner [9,10].

Although previous research evidenced the antioxidant and anti-inflammatory properties of Res in dogs, its role and mechanism in managing canine stress remains to be explored. We hypothesized that Res might alleviate chronic stress in kennel dogs by boosting the immune and antioxidant capacities and regulating the gut microbiome and metabolism. Considering the animal welfare and experimental costs, we were unable to collect gut and brain tissues from dogs to assess the relevant indicators within them. Therefore, we selected mice that were also under chronic stress as a model to obtain evidence in gut and brain tissues, which complemented our findings in dogs. The current study aimed to investigate the stress-relieving effects and underlying mechanism of Res in dogs and mice exposed to chronic stress by combining microbiomics and metabolomics, with a view to provide a reference for the development of alternative strategies for stress management.

## 2. Materials and Methods

### 2.1. Animal Ethics

All experimental procedures were authorized by the Animal Care and Use Committee (approval numbers: 2021E028 and 2022F233) and were performed following the guidelines of the Laboratory Animal Center at South China Agricultural University.

### 2.2. Animal and Housing

#### 2.2.1. Dogs

A total of 18 dogs, around 4 years old, 8 male and 10 females, with an average initial body weight (BW) of 4.80 ± 0.24 kg, were included in this experiment. These dogs had been kept for nutrition experiments in the Qingke Biotechnology Co., Ltd. (Guangzhou, China), for about 3 years. All the dogs were single-housed in kennels (0.70 m × 0.50 m × 0.50 m) located in the same room and had limited access to the outdoor environment or social interaction, except from experimenters. The temperature, humidity, and dark–light cycle of the kennel were not strictly controlled. The volume of ambient sound/noise was 86.38 ± 3.18 decibels. The dogs were offered feed and water ad libitum.

#### 2.2.2. Mice

Three-week old specific-pathogen-free male C57BL/6 mice (Guangdong Medical Laboratory Animal Center, Foshan, China) were housed in the South China Agricultural University animal facility under controlled conditions with the humidity at 50 ± 5% and temperature 25 ± 2 °C in the same room. The mice were offered feed and water ad libitum.

### 2.3. Treatment Administrations and Sample Collection

#### 2.3.1. Experiment 1

The basal diet fed to the dogs was manufactured by the Qingke Biotechnology Co., Ltd. (Guangzhou, China), and met the nutrient recommendations by the Association of American Feed Control Officials (AAFCO, 2017) for dogs. The ingredients and analyzed chemical composition are shown in Appendix A. The dogs were assigned to three treatment groups (*n* = 6/group), namely, the groups of basal diet (Con), basal diet + fluoxetine (Flu), and basal diet + Res (Res), which were balanced for initial BW and sex. After 7 days of acclimation, the dogs in the Con group were fed the basal diet and an empty capsule made of starch daily throughout the experimental period, while the dogs in the Res and Flu groups were fed the basal diet and starch capsule containing 100 mg/kg BW Res [11] and 1 mg/kg BW fluoxetine [12], respectively, based on previous research. The dosage was calculated based on the conversion of the body surface area between mice and dogs. The timeline and sampling points of this experiment are shown in Figure 1a.

T1 and T2 represent the sampling time points at the beginning and end of experiment 1, respectively. After 8 hours of fasting, 6 mL of blood was sampled from each dog from the forelimb vein into a blood-collecting tube at T1 and T2. The tubes were left to tilt for 30 min and then centrifuged at 3500 rpm at room temperature for 15 min. The supernatant was aliquoted into microcentrifuge tubes and stored at −80 °C for further analyses. Fresh feces were also collected at T1 and T2 and stored at −80 °C for further analyses.

The fecal microbiota of dogs was cultured in vitro while referring to an established in vitro colonic incubation approach [13]. The fermentation substrate was 1 g of the basal diet of dogs. Different doses of Res were added into the medium so that the concentrations of Res in the media of the Con, L, M, and H groups (*n* = 6/group) were 0, 0.125, 0.25, and 0.5 mg/mL, respectively. Fresh feces from dogs were mixed with phosphate-buffered saline, which were then filtered to obtain the bacterial solution. A total of 7 mL of bacterial solution was added to each medium and incubated anaerobically for 72 h. The pH of the fermentation broth was measured at 0, 6, 12, 48, and 72 of incubation and 2 mL of fermentation broth was collected for further analysis.

#### 2.3.2. Experiment 2

As shown in Figure 1b, the mice were randomly assigned into 5 groups (*n* = 8/group) following acclimation for one week: the control group (Con), chronic unpredictable mild stress (CUMS) group (CUMS), CUMS + 100 mg/kg BW Res (L), CUMS + 200 mg/kg BW Res (M), and CUMS + 300 mg/kg BW Res (H). All mice except the Con group were subjected to CUMS. Mice in the Con and CUMS groups were treated with a 10 mL/kg 1% carboxymethylcellulose (CMC) suspension by oral gavage daily. Meanwhile, mice in the other three groups were administered 100, 200, and 300 mg/kg BW Res in 1% CMC, respectively [11]. The CUMS procedures were conducted as described previously, including 45° cage tilting for 12 h, tail pinching for 5 min, cage shaking for 10 min, damp bedding for 12 h, and restraint in a tube for 2 h [14]. The timeline and sampling points of this experiment are shown in Figure 1b. At the end of the experiment, the mice were sacrificed by cervical dislocations after being fasted for 12 h. The whole-brain and colon tissues were collected and stored at −80 °C for further analysis.

### 2.4. Behavior and Sample Analyses

#### 2.4.1. Behavior Assessments in Dogs and Mice

The open-field test (OFT) was performed at T1 and T2 based on a validated method [15] to evaluate the stress levels of dogs. Specifically, the dogs were alternately placed in an empty area (5.20 m × 4.80 m) and were given 3 min for adaptation to the novel environment and another 3 min for behavioral recording with cameras. One trained researcher coded the videos collected in the OFT and recorded the frequency or duration of the behaviors indicative of dog stress or fear responses, as listed in Appendix A [16].

The OFT was also used to measure the level of stress in the mice. Specifically, the mice were placed in a white open box with a camera positioned directly above the center. The bottom area of the box (45 cm × 45 cm × 45 cm) was evenly divided into 16 squares, the four central squares were considered as the central region, and the rest were the peripheral region. A video-tracking system was applied to record the total distance the mice traveled in the two regions to evaluate the anxiety of the mice. The forced swim test (FST) could assess the anxiety and depression mood in mice [17]. The mice were placed individually in a glass beaker containing water (26 ± 1 °C). After 2 min of adaptation, the time the mice swam in the water was recorded.

#### 2.4.2. Serum and Brain Biochemistry Analyses in Dogs and Mice

The total antioxidative capacity (T-AOC) and malondialdehyde (MDA) in the serum of dogs were determined using commercial kits (Nanjing Jiancheng Bioengineering Institute, Nanjing, China) following the manufacturer’s protocols. Serum 5-HT, BDNF, CRH, ACTH, cortisol, interleukin 10 (IL-10), and immunoglobulin A (IgA) in the serum of dogs were evaluated using a commercial canine enzyme-linked immune-sorbent assay (ELISA) kits (MEIMIAN, Jiangsu Meimian Industrial Co., Ltd., Yancheng, China). Trp and 5-HT in the whole brains of mice were also measured using a mouse ELISA kit.

#### 2.4.3. Gene Expression Analysis in Mice

The tissue total RNA was isolated with an RNA Pure tissue kit from EZBioscience Co., Ltd. (Roseville, CA, USA). For real-time PCR analysis, reverse transcription was performed using an A0010CGQ reverse transcription kit from EZBioscience Co., Ltd. A SYBR Green qPCR SuperMix was performed on an ABI 7500 real-time PCR system according to the manufacturer’s instructions. The values were normalized against control NADPH. The relative mRNA expression levels were calculated using the 2^−ΔΔCt^ method. The sequences of the real-time PCR primers used are listed in Appendix A.

#### 2.4.4. Fecal SCFAs and Branched-Chain Fatty Acids (BCFAs) Analysis in Dogs

A total of 1 mL of ultra-pure water was added to 0.2 g feces, which was vortexed for 5 min. The mixture was placed in an ice bath for 10 min of ultrasonic crushing and centrifuged at 13,000 rpm for 10 min at 4 °C, whereafter 20 μL of 25% metaphosphoric acid solution and 0.25 g anhydrous sodium sulfate were added to the supernatant and then mixed for 2 min. Each sample was added with 1 mL of methyl tert-butyl ether, vortexed for 5 min, and centrifuged at 13,000 rpm for 5 min at 4 °C. The supernatant was harvested and filtered through a 0.22 μm Millipore pore membrane filter to a sample vial. The quantitative analysis of SCFAs and BCFAs of the samples were carried out using a gas chromatography-MS-QP2020 system (Shimadzu, Tokyo, Japan) following the method used in our laboratory [18].

#### 2.4.5. Feces Microbiota Analysis in Dogs

The DNA from fresh fecal samples was extracted using the CTAB according to the manufacturer’s instructions. The V3–V4 region of the bacterial 16S rRNA gene was amplified using the primers 341 F (5′-CCTACGGGNGGCWGCAG-3′) and 805 R (5′-GACTACHVGGGTATCTAATCC-3′) with a barcode. PCR amplification was performed in a total volume of 25 μL reaction mixture containing 25 ng of template DNA, 12.5 μL PCR Premix, and 2.5 μL of each primer. The cycling parameters were 98 °C for 30 s, followed by 32 cycles of 98 °C for 10 s, 54 °C for 30 s, and 72 °C for 45 s, with a final extension at 72 °C for 10 min. The PCR products of each sample were detected by 2% agarose gel electrophoresis, purified by AMPure XT beads (Beckman Coulter Genomics, Danvers, MA, USA), quantified by Qubit (Invitrogen, Waltham, MA, USA), and sequenced on a NovaSeq PE250 platform (LC-Bio Technology Co., Ltd., Hang Zhou, Zhejiang Province, China). The reads were paired using FLASH. The quality filtering on the raw reads was performed to obtain the high-quality clean tags using fqtrim (version 0.94). Chimeric sequences were filtered using Vsearch software (version 2.3.4). After dereplication using DADA2 (version 1.26.0) in the QIIME2 software (version 2023.5), the feature table and feature sequence were obtained. The feature abundance was then normalized according to the SILVA (release 138) classifier using the relative abundance of each sample.

The α-diversity (Simpson index) was calculated in QIIME2, and the principal component analysis (PCA) results were displayed using the SIMCA-P 14.1 software. The nsegata-LEfSe software (version 1.1.2) was used to perform Linear discriminant analysis Effect Size (LEfSe) analysis and the score of linear discriminant analysis (LDA) was set at ≥4. The Mann–Whitney U test was applied to examine the differences in the relative abundance of microbiota between groups.

#### 2.4.6. Untargeted Serum and Fecal Metabolome Analyses in Dogs

A total of 200 μL of serum sample was mixed with 800 μL of methanol. After two min of vortexing, the mixture was centrifuged at 14,500 rpm for 15 min at 4 °C. The supernatant was blow-dried with nitrogen and then dissolved with 200 μL of methanol. The samples were ultrasonically processed in an ice bath for 10 min and then centrifuged at 14,500 rpm for 15 min at 4 °C. Supernatant were filtered through a 0.22 μm Millipore pore membrane filter to a sample vial for UPLC-Orbitrap-MS/MS analysis [19]. The process of preparation of feces and fermentation broth for UPLC-Orbitrap-MS/MS analysis were similar to the serum samples with a minor modification. The Compound Discoverer 2.1 (Thermo Fisher Scientific, Waltham, MA, USA) data analysis tool was applied to complete the raw data pre-processing and was applied to identify the metabolites by searching the mzCloud library and mzVault library. The PCA was performed with SIMCA-P 14.1 software. The differential metabolites were screened out and a KEGG pathways enrichment analysis of the differential metabolites was performed using MetaboAnalyst 5.0 (https://www.metaboanalyst.ca), accessed on 10 March 2023.

#### 2.4.7. Targeted Metabolome Analyses of Serum, Feces, and Broth Fermentation in Experiment 1

Targeted metabolomics was applied to quantify the metabolites involved in the Trp metabolism (Appendix A) and neurotransmitters, including γ-aminobutyric acid (GABA), glutamate (Glu), acetylcholine (ACh), and dopamine. Standard compounds were dissolved with methanol–water (1:1) and then diluted in a gradient to a standard solution (i.e., 1000, 500, 100, 10, 1, 0.5, 0.1, 0.01, 0.001 ng/mL). The standard solution was further analyzed to build standard curves using UPLC-Orbitrap-MS/MS while referring to the method in [20]. Finally, the raw data were imported into the Xcalibur software (version 4.3) to quantify the concentration of metabolites according to their standard curves.

#### 2.4.8. MetOrigin Analysis for Gut Microbiota and Microbial Metabolic Pathways in Dogs

MetOrigin, a bioinformatics tool that can trace the biological origins of microbial metabolites and related metabolic pathways (http://metorigin.met-bioinformatics.cn/, accessed on 10 March 2023) [21], was utilized to analyze and visualize the biological and statistical correlation between differential microbiota and metabolic pathways, accessed on 5 August 2023.

### 2.5. Statistical Analysis

SPSS 26.0, GraphPad Prism 8.0, and Adobe Illustrator were used for the statistical analysis and graphic presentation. For independent samples, Student’s *t*-test and one-way analysis of variance with the LSD multiple comparisons test were performed to compare the differences between groups. A two-way repeated measure analysis of variance with Bonferroni adjustment for multiple comparisons was performed to analyze the differences between the groups (i.e., treatment administrations (TRTs)) at different time points (i.e., time). The data are presented as the mean ± SEM. Significant differences were set at *p* < 0.05 and tendencies at 0.05 ≤ *p* < 0.10.

## 3. Results

### 3.1. Experiment 1

#### 3.1.1. Behaviors Related to Fear and Anxiety in the OFT in Dogs

The main effect of time on the escape attempt was significant, that is, the total duration of the escape attempt of dogs decreased over time (*p* = 0.04, Appendix A). The effect of time × TRT was significant on the duration of the pace and exploratory behavior (Appendix A). Specifically, the length of time spent on pacing in the Res group was significantly lower at T2 than T1 (*p* = 0.03, Appendix A), whereas the dogs in the Res group tended to spend more time exploring at T2 than T1 (*p* = 0.08, Appendix A).

#### 3.1.2. Serum 5-HT, BDNF, and Hormones of HPA Axis in Dogs

As shown in Figure 2a, the serum level of 5-HT in the Res and Flu groups increased at T2 compared with T1 (*p* = 0.002 and *p* = 0.01), while there was a significant decrease in the Con group. The serum concentration of 5-HT in the Con group was lower than the Flu and Res groups by the end of the experiment (*p* = 0.09 and *p* = 0.004, Figure 2a). Both the Flu and Res treatments significantly elevated the serum BDNF by the end of the experiment (T2) compared with the control and T1 (Figure 2b). The concentrations of CRH and ACTH in the serum descended markedly during the experimental period in the Res group (*p* = 0.01, Figure 2c; *p* = 0.03, Figure 2d). The level of cortisol was lower at T2 compared with T1 in the Flu group (*p* < 0.001) and Res group (*p* = 0.02) (Figure 2e).

#### 3.1.3. Serum GABA, ACh, Glu, and Dopamine in Dogs

Time and TRT did not exert obvious influences on the concentrations of GABA, ACh, and Glu in the serum (Appendix A). However, dopamine in all groups was significantly increased at T2 relative to T1 (*p* < 0.001, Appendix A).

#### 3.1.4. Cytokine, IgA, and Antioxidant Measures in Dogs

The concentration of IL-10 in the Flu group was relatively lower than in the Con and Res groups at T1 (*p* = 0.01 and *p* = 0.07, Figure 3a). But IL-10 in the Flu and Res groups was at a higher level than in the Con group at T2 (*p* = 0.008 and *p* < 0.001, Figure 3a), which was attributed to the increase in IL-10 in these two groups over time (*p* < 0.001 and *p* = 0.001, Figure 3a). Similarly, there was a considerable increase in the IgA concentration in the Flu and Res groups at T2 compared with that at T1 (*p* < 0.001 and *p* = 0.06, Figure 3b), which resulted in a lower IgA in the Con group than the Flu and Res groups at T2 (*p* = 0.004 and *p* = 0.01, Figure 3b). As for T-AOC, its concentration in the Res group was higher than in the Con group (*p* = 0.03) and Flu group (*p* = 0.06) at T2 (Figure 3c), mainly due to its reduction in the Con group (*p* < 0.001) and rise in the Res group (*p* = 0.02) over time (Figure 3c). And only the Con group manifested a noticeable increase in the MDA level throughout the experiment (*p* = 0.002, Figure 3d).

#### 3.1.5. Gut Microbiota and Fecal SCFAs and BCFAs in Dogs

The Simpson index was higher in the Con group than the Res group at T2 (*p* = 0.016, Figure 4a). And the PCA plot displayed obvious separation between the Con and Res groups (Figure 4b). The ratio of abundance of Firmicutes to Bacteroidota was significantly higher in the Res group than the Con group (Figure 4d). There was a relatively higher abundance of *Holdemanella* and less abundance of *Faecalibacterium* in the Res group than the Con group (Figure 4e). The Mann–Whitney U test suggested that *Alloprevotella* and *Lachnospira* were enriched in the Con group, while the resveratrol administration increased the abundance of *Blautia* and *Holdemanella* (Figure 4f). As displayed in Figure 4g, *Erysipelotrichales*, *Holdemanella_unclassified*, *Holdemanella*, and *Erysipelotrichaceae* displayed significant enrichment in the Res group, while *Oscillospirales*, *Ruminococcaceae*, *Faecalibacterium*, and *Clostridia* were enriched in the Con group.

Butyric acid had an obvious increase in the Res group over the experimental period (*p* = 0.04, Figure 4h). As shown in Figure 4h, fecal isovaleric acid in the Res group ascended over time (*p* = 0.008), which resulted in the Res group having a slightly higher level of isovaleric acid at T2 than the control (*p* = 0.06). The levels of acetic and isobutyric acid in the feces of dogs in both groups were elevated at T2 compared with T1 (*p = 0.07* and *p* = 0.04, Figure 4h).

#### 3.1.6. Metabolome in Dogs

The PCA score plot indicated there was an evident separation of the serum and feces metabolites (Appendix A) between the Con and Res groups at T2. As shown in Appendix A, the Res group had higher concentrations of astringin and scopoletin and a lower concentration of indoxyl sulfate in the serum. The resveratrol increased the content of benzofuran and resveratrol in feces (Appendix A). To investigate the changes of metabolic processes, the KEGG pathway enrichment analysis of differential metabolites in serum and feces were applied with thresholds of |log2 (FC)| > 1 and VIP > 1 (Appendix A). The affected metabolic pathways in the serum were arginine and proline metabolism, metabolism of α-linolenic acid and arachidonic acid, biosynthesis of unsaturated fatty acids, pantothenate and CoA biosynthesis, and glycan biosynthesis and metabolism (i.e., glycosphingolipid biosynthesis; Appendix A). Furthermore, resveratrol supplementation also changed the metabolic pathways in the feces, including the biosynthesis of phenylalanine, tyrosine, and Trp; the metabolism of tyrosine and Trp; and the purine metabolism (Appendix A).

The resveratrol administration had a tendency to increase the serum Trp concentration (*p* = 0.07, Figure 5a). The Res group had a higher 5-HT level than the Con group at the end of the trial (*p* = 0.002, Figure 5a). Conversely, picolinic acid (PA) increased significantly in the two groups with time but was higher in the Con group at T2 (*p* = 0.08, Figure 5a). Numerically, indole in the Res group decreased over time (*p* = 0.133, Figure 5a), while it remained relatively stable in the Con group. And Figure 5b provides a comprehensive view of the changes in the metabolites in the Trp pathway in the serum.

Similar to the trends of the serum Trp, the Res group had a higher Trp level in the feces than the Con group at T2 (*p* = 0.002, Figure 5c). Remarkably, the concentration of 5-hydroxytryptophan (5-HTP) in the Con group tended to decrease with time (*p* = 0.07, Figure 5c), but did not change much in the Res group, which resulted in a significantly higher concentration of 5-HTP in the Res group than the Con group at T2 (Figure 5c). Resveratrol elevated the concentrations of melatonin (MLT) (*p* = 0.04), PA (*p* = 0.005), indole acrylic acid (IA) (*p* < 0.001), and indole-3-acetic acid (IPA) (*p* = 0.002) over the experimental period (Figure 5c). The detailed variation in the metabolites related to the Trp metabolism in the feces is presented in Figure 5d.

#### 3.1.7. Biological Relationship of Gut Trp Metabolism and Microbes in Dogs

A MetOrigin analysis was used to find the fecal microbiota at the genus level that were biologically relevant to Trp metabolism in the gut at T2. As shown in the Sankey network (Appendix A), the Res group had higher relative abundances of *Haemophilus*, *Klebsiella*, *Desulfovibrio*, *Alistipes*, and *Lachnoanaerobaculum*, which had a positively biological correlation with the level of Trp and had a negatively biological correlation with indole in the feces of the dogs. Following the resveratrol administration, the abundances of gut microbiota, such as *Fusobacterium* and *Clostridium*, that were positively correlated with indole and negatively correlated with Trp were downregulated. The quantitative detection of Trp and indole in feces at T2 showed that the level of Trp was higher in the Res group compared with that in the Con group (*p* = 0.002, Appendix A), while the concentration of indole in the Res group was numerically lower than that in the Con group (Appendix A).

#### 3.1.8. pH and Trp Metabolism in Fermentation Broth of Experiment 1

The effect of time × TRT was significant for pH, which decreased over time during the broth fermentation (*p* = 0.001, Figure 6a). As shown in Figure 6a, there was no significant difference in the pH between all the groups at 0 h. At 6 h, the pH of the M group was lower than that of the Con group (*p* = 0.064). The pH of the H group was significantly lower than the Con group (*p* = 0.04) at 12 h. At 24 h, the pHs of the M and H groups were lower than that of the Con group (*p* = 0.03 and *p* = 0.08). At 48 h, the pH of group H was significantly lower than that of the Con group (*p* = 0.008), while there was no significant difference in the pH between the groups at 72 h.

The pH of the fermentation broth reflected the microbial fermentation process. According to the change in the pH, we believe that the Res had the greatest influence on the microbial fermentation at 48 h, and we detected the metabolites of the Trp metabolism in the fermentation broth at this time. The levels of Trp in the fermentation broth of the M and H groups were higher than that in the Con group (*p* = 0.02 and *p* = 0.05, Figure 6b). The Con group had a significantly higher concentration of tryptamine (TAM) than the other groups (*p* = 0.01, *p* = 0.009, and *p* < 0.001; Figure 6b). Similarly, Res reduced the levels of indole ethanol (IE) in the fermentation broth (*p* = 0.10, *p* = 0.06, and *p* = 0.01; Figure 6b). As shown in Figure 6b, the levels of IPA in the M and H groups were lower than in the Con group (*p* = 0.09 and *p* = 0.01). The detailed variation in the metabolites related to the Trp metabolism in the fermentation broth is presented in Figure 6c.

### 3.2. Experiment 2

Experiment 1 found that resveratrol had the potential to relieve stress in the dogs, and this effect might be attributed to changes in the tryptophan metabolism related to gut microbiota, especially the levels of tryptophan in the hindgut and blood. However, the way by which the changes in the tryptophan levels related to the microbiota in the hindgut influenced the levels of tryptophan in the blood is still not well understood. Given the unavailability of brain and hindgut tissue samples from dogs, we employed mice under chronic stress to explore the transportation of tryptophan from the hindgut into the peripheral circulation, as well as the levels of tryptophan and 5-HT in the CNS.

#### 3.2.1. Behavioral Tests in Mice

In the OFT, the total distance traveled in the peripheral region by the mice in the CUMS group was shorter than in the Con group (*p* = 0.04, Appendix A), whereas Res could increase the distances traveled by the mice in the peripheral region (*p* = 0.005, *p* = 0.01, and *p* = 0.05; Appendix A). The distance traveled by the mice in the CUMS group in the central region was shorter than the other groups (*p* = 0.02, *p* = 0.001, *p* = 0.001, and *p* = 0.02; Appendix A). And the trajectory of the mice in the OFT is displayed in Appendix A. In the FST, the duration of swimming of the mice in the CUMS group was less than the other groups (*p* < 0.001, *p* = 0.02, *p* < 0.001, and *p* < 0.001; Appendix A).

#### 3.2.2. Trp and 5-HT in the Whole Brains of Mice

The CUMS decreased the level of Trp in the whole brains in mice (*p* = 0.008, Figure 7a), while the oral intake of Res could restore the levels of Trp (*p* = 0.02, *p* = 0.008, and *p* < 0.001; Figure 7a). Similarly, the concentration of 5-HT in the CUMS group was lower than the other groups (*p* = 0.05, *p* = 0.01, *p* = 0.002, and *p* < 0.001; Figure 7b).

#### 3.2.3. Expression of Genes Related to the Tight Junction Protein, aryl hydrocarbon Receptor (AhR), and Trp Transporters in the Colons of Mice

As demonstrated in Figure 8a, the CUMS decreased the relative mRNA expression of *claudin-1* in the colon compared with the Con group (*p* < 0.001), while 100 and 200 mg/kg BW Res restored the *claudin-1* mRNA expression (*p* < 0.001 and *p* < 0.001). The gene expression of *CYP1A1* in the colon was upregulated in the L, M, and H groups compared with the CUMS group (*p* = 0.001, *p* < 0.001, and *p* = 0.08; Figure 8b). But, neither the CUMS nor the Res treatment had a significant effect on the *CYP1B1* (Figure 8c). Compared with the Con group, the mRNA expressions of *SLC7A8*, *SLC16A10*, and *SLC6A19* in the colon were decreased. After receiving the administration of Res, the gene expressions of *SLC7A8*, *SLC16A10*, and *SLC6A19* were recovered in the L, M, and H groups (*p* < 0.001, *p* = 0.001, and *p* < 0.001; Figure 8d) (*p* < 0.001, *p* < 0.001, and *p* = 0.04; Figure 8e) (*p* < 0.001, *p* < 0.001, and *p* = 0.04; Figure 8f).

## 4. Discussion

The OFT and FST are commonly used to assess stress and despair in rodents. Mice that suffer from stress display decreased exploration in the central region in the OFT, and exhibit a shorter swimming time in the FST [22]. Different doses of Res increased the travelling distance and swimming time of the mice, indicating that Res could improve the stress and anxiety-like behaviors in our study. In a previous study, Res was shown to reduce the depressive-like behavior caused by chronic stress in rats [23]. Our study also found that resveratrol reduced pacing, which is a typical behavior indicative of anxiety. Explorative behavior, as a positive reaction to a novel environment, is usually reduced in dogs experiencing mental stress. Therefore, the increased exploratory behavior in the Res group suggested that resveratrol could ameliorate the stress of dogs in the OFT.

### 4.1. Neurotransmitters and Hormones of HPA Axis

Although 5-HT cannot cross the blood–brain barrier (BBB), there is evidence indicating that the content of 5-HT in the brain correlates positively with that in the blood. Studies on dogs found that a lower concentration of serum 5-HT was associated with increased aggressive behavior and impulsivity [24]. Our results indicate that resveratrol had similar effects to fluoxetine, a selective 5-HT reuptake inhibitor, in increasing serum 5-HT. In mice, resveratrol has also been reported to ameliorate the mild stress caused by unpredictable stimuli by enhancing the 5-HT content in the hippocampus [25]. Peripheral concentrations of other neurotransmitters (e.g., GABA, dopamine, Ach, and Glu) were reported to indirectly reflect mental disorders. In our study, resveratrol had no obvious impact on these neurotransmitters. Therefore, the therapeutic target of resveratrol on canine chronic stress in the present experiment was more likely to be 5-HT rather than other neurotransmitters. The dogs with anxiety had a lower concentration of serum BDNF, which could cross the BBB and act as a neurotrophin to promote the expression of rate-limiting enzymes in the 5-HT biosynthesis process. Niu et al. (2020) reported that resveratrol could restore the decreased serum BDNF in a rat model of schizophrenia [26]. The increased serum BDNF might have partially mediated the anti-stress effects of resveratrol on the dogs in the Res group.

Chronic stress can lead to the dysregulation of the HPA axis, compromised immune function, and an increase in the chance of chronic diseases. Previous studies showed that resveratrol could alleviate anxiety-like behavior by downregulating HPA axis hyperactivity in rats [27]. Our study also found that resveratrol could reduce the concentration of hormones in the HPA axis. Accordingly, the effect of resveratrol in relieving canine chronic stress in our study was partly attributed to the fact that resveratrol could normalize the function of the HPA axis.

### 4.2. Immune Function and Oxidative Stress

The resveratrol increased the serum IgA and IL-10 concentrations in the dogs compared with the control. We hypothesized that the resveratrol treatment could reduce the susceptibility of dogs to pathogenic microorganisms by enhancing the serum IgA and IL-10 to mitigate the excessive development of inflammation, thereby improving the immune status in dogs under stress [28].

External pressure can lead to the production of excessive oxidative free radicals that can impair neurons and induce anxiety and depression. Our study provided evidence that resveratrol alleviated the oxidative stress in the dogs by increasing T-AOC, which represents the total capacity of various antioxidant systems [29].

### 4.3. Gut Microbiota, SCFAs, and BCFAs

Current evidence indicates that gut microbiota might regulate mental stress through the MGB axis, which is a bidirectional communication system between the gut and brain. Resveratrol was shown to modulate the structure of the gut microbiota and reduced the microbial diversity in our study. Huang et al. (2022) has also reported a decrease in α-diversity in stressed rats after treatments with probiotics [30], which is possibly due to the prevention of the proliferation of certain pathogenic bacteria by the treatments. The improvement of stress-related diseases is associated with an increase in the ratio of Firmicutes and Bacteroidota, and the former could contribute to the intestinal integrity and barrier [31]. The intercellular tight junction structure is an important part of the intestinal barrier and is composed of claudin-1 and other tight junction proteins. The decreased gene expression of *claudin-1* indicated the disruption of the intestinal barrier [32]. A total of 100 and 200 mg/kg BW Res could restore the CUMS-induced damaged intestinal barrier by upregulating the gene expression of claudin-1. We hypothesized that the function of Res to protect the intestinal barrier may be related to the regulation of the abundance of firmicutes in the gut by Res. We hypothesized that the abundance of Firmicutes increased by resveratrol had a positive effect on the intestinal barrier in stressed dogs.

As fermentation products of gut microorganisms, SCFAs can regulate depressive symptoms by inhibiting neuroinflammation and acting on receptors in the CNS through the BBB. A study suggested that resveratrol has the potential to increase the intestinal content of butyric acid in mice suffering from weaning stress [33]. Resveratrol increased the abundance of *Blautia* and *Holdemanella* in our study, which are butyric-acid-producing bacteria, and the resultant increase in intestinal butyric acid was shown in humans and rodents to improve mental disorders by increasing the 5-HT concentration and upregulating the BDNF expression in the CNS [34]. The current study found increased fecal butyric acid and serum 5-HT and BDNF in the dogs of the Res group, which supported our speculation above. Function analyses of the microbiota biomarkers found that the Res group was enriched with *Erysipelotrichales* and *Holdemanella*, which also were identified as butyric acid producers in the gut, whereas *Fusobacterium*, a proinflammatory pathogen that can cause colonic inflammation [35], was more abundant in the Con group. Studies highlighted that intestinal inflammation could lead to gut barrier damage, allowing the diffusion of endotoxin to the CNS, which could impair mental health as a result of neurotoxins and the hindered function of neurotransmitters. Collectively, our results presented supportive evidence that resveratrol could promote the proliferation of butyric-acid-producing bacteria and suppress the growth of harmful bacteria associated with intestinal inflammation to ameliorate stress in the dogs through the MGB axis.

Generally, BCFAs were considered as metabolites from protein fermentation that were detrimental to host health. However, a recent report showed that intestinal isovaleric acid in chronically stressed rats had a lower concentration than in normal rats, but was elevated after receiving the antidepressant treatment [36]. Consistently, the treatment with resveratrol also increased the fecal isovaleric acid in the dogs, indicating its potential role in stress management, while the specific mechanism needs to be elucidated in the future.

### 4.4. Metabolome in Serum, Feces, and Fermentation Broth

Our experiment found that resveratrol regulated various metabolites and metabolic processes in the dogs. Specifically, resveratrol could increase the serum concentrations of astringin and scopoletin, which have potent bioactivity in scavenging free radicals and inhibiting inflammation. It is worth noting that scopoletin might exert antidepressant effects that are related to the serotonergic system [37]. Compared with the Res group, the Con dogs showed a higher concentration of serum indoxyl sulfate, a toxin from indole sulfonation that could increase the permeability of BBB and aggravate stressful and anxious symptoms. Similar findings have been reported stating that resveratrol reduced the accumulation of indoxyl sulfate in rats that experienced acute kidney injury [38].

Metabolomics analyses targeting therapies for mental diseases discovered that the disturbance of the metabolism of amino acids, lipids, and energy was closely associated with chronic stress. In accordance with previous findings, our results of differential metabolic pathways revealed that the resveratrol treatment was accompanied with the changes in arginine and proline metabolism, α-linolenic acid metabolism, arachidonic acid metabolism, biosynthesis of unsaturated fatty acids, pantothenate and CoA biosynthesis, and glycan biosynthesis and metabolism in the serum. Nitric oxide, a messenger molecule from arginine catabolism, which affects synaptic plasticity, was suggested to be involved in the mechanism by which arginine and proline metabolism participates in the regulation of mental stress. As the precursor of unsaturated fatty acids in vivo, such as eicosapentaenoic acid (EPA) and docosahexaenoic acid (DHA), α-linolenic acid was suggested to exert neuroprotective effects by ameliorating oxidative stress. And EPA and DHA could improve emotional health by increasing the concentration of monoamines and BDNF in the brain [39]. Our results also show that the serum 5-HT and BDNF levels were upregulated by resveratrol, and the changes in the linolenic acid metabolism might have been one of the key mechanisms of resveratrol in relieving the canine stress. Fan et al. (2021) reported that arachidonic acid might be a key metabolite that impacts synaptic plasticity and exacerbates depression [40]. Meanwhile, prostaglandin (PG) E2 and PGD2 in arachidonic acid metabolism also modulates synaptic transmission and plasticity. Resveratrol has been reported to exert its anti-inflammatory function through disrupting the metabolism of arachidonic acid [41], and our study provided additional evidence that resveratrol targets arachidonic acid metabolism to reduce inflammation and relieve stress in the dogs. One possible role of pantothenate and CoA biosynthesis in stress management is that its inhibition of pantothenate and CoA metabolism could downregulate oxidative stress. The disruption of glycosphingolipid metabolism was observed in a mouse model of autism [42], while the normalization of glycosphingolipid biosynthesis is critical for neuronal functions. Overall, the effects of resveratrol treatment on serum metabolites and metabolic pathways suggested that resveratrol shows potential anti-stress effects, which, from the perspective of metabolism, were significantly likely to occur through ameliorating oxidative stress and inflammation and promoting the expressions of 5-HT and BDNF.

The disturbance of amino acid metabolism detected in feces was of particular relevance to the fecal metabolome of individuals with chronic stress. Similar to previous studies [43], we also observed that the differential metabolic pathways in feces relevant to reduced stress were enriched in amino acid metabolism, such as phenylalanine, tyrosine, and Trp biosynthesis, and the metabolism of tyrosine and Trp. We hypothesized that Trp metabolism might be an important therapeutic target of resveratrol for alleviating stress in dogs. Trp is mainly catabolized via three pathways, namely, the 5-HT and kynurenine pathways in the host, and the indole pathway mediated through bacteria in the intestine. Trp metabolism was suggested to be closely associated with chronic stress, and was reported to be disturbed in fearful dogs [44]. The overactivation of kynurenine acid pathway triggered by stress not only resulted in the accumulation of quinolinic acid (QA), a neurotoxic metabolite, but also led to the excessive depletion of peripheral Trp. A study on depression also found that a Trp-rich diet could restore the imbalance between the kynurenine pathway and 5-HT pathway to alleviate anxiety-like behaviors in mice models [14]. The quantification of metabolites in the Trp metabolism revealed that resveratrol regulated Trp and its metabolites in both the serum and feces in the dogs. There was a significant increase in the serum 5-HT by the resveratrol treatment, while the serum PA in the kynurenine acid pathway decreased upon the resveratrol administration in our study. Due to limitations with the detection method, QA was not detected in the serum samples. Taken together, we suggest that resveratrol can attenuate the overactivation of the kynurenine pathway caused by stress in dogs and increase the circulating Trp to enhance the synthesis of 5-HT in the brain.

Gut microbes can break down Trp to produce metabolites, such as indole and tryptamine. The increase in the fecal Trp by resveratrol in the current study may be attributed to the reduced abundance of certain bacteria that can utilize Trp in the gut, which was confirmed by the reduced consumption of Trp and reduction in the microbial Trp catabolites, like TAM and IE, in the in vitro fermentation [20]. The MetOrigin analysis revealed that *Fusobacterium* and *Clostridium* might be the candidate bacteria that catabolized intestinal Trp, which are positively correlated with generalized anxiety disorder [45] but were shown to be inhibited by resveratrol. On the other hand, some gut bacteria were able to increase the serum Trp concentration [46]. Positive correlations were identified between fecal Trp and bacteria, such as *Haemophilus*, *Klebsiella*, *Desulfovibrio*, *Alistipes*, and *Lachnoanaerobaculum.* The resveratrol intake might provide a favorable intestinal condition for the growth of these bacteria, which might promote the synthesis of Trp in the intestine. Borisova et al. (2020) reported that fucose could normalize the blood Trp level and ameliorate behavioral disorders by restoring the growth of Trp-producing bacteria in the gut [46]. Gao et al. (2019) also suggested that the circulating level of amino acids might be partially dependent on the alternations of amino acid metabolism in the gut affected by microbes [10]. Considering that Res increased the level of Trp in the serum of dogs, we speculated that resveratrol could modulate the population and diversity of gut bacteria involved in Trp metabolism to increase the content of Trp in the hindgut and allow for more Trp be transferred by Trp transporters in the colon to the blood and brain and converted to 5-HT [10]. To verify the above speculation, we measured the gene expressions of Trp transporters, including *SLC7A8*, *SLC16A10*, and *SLC6A19*, in the colons of mice [47,48]. We found that CUMS decreased the mRNA expressions of Trp transporters, which might have been associated with the damage of the intestinal epithelial structural induced by CUMS [49]. The fact that Res restored the expressions of the above genes in our study suggests that Res could increase the Trp levels in the hindgut by modulating the gut microbes and Trp transporters, thus allowing for more Trp to be transported from the hindgut to the peripheral circulation and the brain for the host synthesis of 5-HT.

Due to the limitation of the detection methods in our study, only a fraction of microbial Trp catabolites were quantified, where they did not seem to be downregulated by resveratrol, and there might have been catabolites affected by resveratrol that remained undetected among the metabolites of specific microbes. Surprisingly, resveratrol upregulated the concentrations of IA and IPA in the gut, but not in the serum. Previous studies illustrated that IA and IPA participate in immune regulation and the maintenance of the intestinal barrier by activating AhR, and *CYP1A1* and *CYP1B1* are target genes regulated by AhR [50]. In this study, we found that Res upregulated the gene expression of *CYP1A1* in the colons of CUMS-exposed mice, which suggests that the AhR signaling pathway was activated. Considering that IA and IPA can act as ligands for AhR, we believe that Res can activate the AhR/CYP1A1 pathway by promoting the production of IA and IPA in the colon, which played an important role in improving the gut barrier and alleviating inflammation, thus relieving chronic stress [51,52,53,54]. In summary, changes in metabolism, especially with Trp, were important underlying mechanisms of resveratrol actions in promoting intestinal health and relieving canine stress.

### 4.5. Limitations and Future Directions

Due to the limitation of experimental animals, dogs under normal living conditions were not included in this study as a negative control. Although we identified several bacteria that might affect canine stress by participating in the modulation of Trp metabolism in the gut, further in vitro cultivation and transplantation of candidate bacteria will be needed to evaluate their roles in contributing to the improvement of mental stress in dogs. Additionally, the use of total RNA-seq technology might help to identify new gene targets for the stress-relieving effects of resveratrol.

## 5. Conclusions

Our study demonstrated that dietary supplementation with Res could ameliorate environmental chronic stress by increasing the 5-HT and BDNF levels, improving the antioxidant capacity and immune function, and restoring abnormalities of the HPA axis. The stress-relieving effects of resveratrol might be partially attributed to its role in modulating Trp metabolism and restructuring the gut microbiome in the present experiment (Appendix A).

## Figures and Tables

**Figure 1 antioxidants-14-00195-f001:**
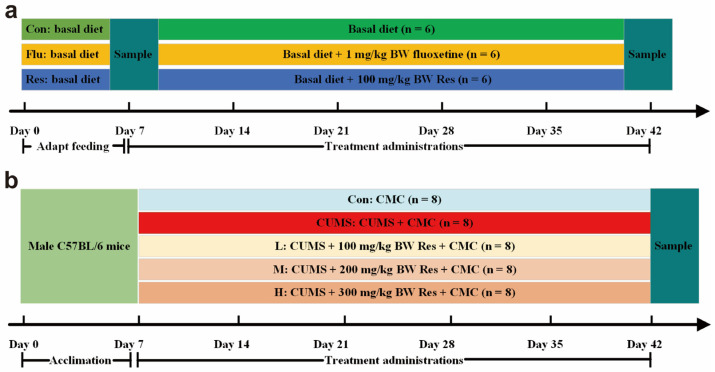
Timeline and sampling time points. (**a**) Basal diet (Con), basal diet + fluoxetine (Flu), and basal diet + Res (Res), where T1 and T2 represent the sampling time points at the beginning and end of the experiment 1; (**b**) CMC (Con), CUMS + CMC group (CUMS), CUMS + 100 mg/kg BW Res (L), CUMS + 200 mg/kg BW Res (M), and CUMS + 300 mg/kg BW Res (H). Res, resveratrol; CMC, carboxymethylcellulose; CUMS, chronic unpredictable mild stress.

**Figure 2 antioxidants-14-00195-f002:**
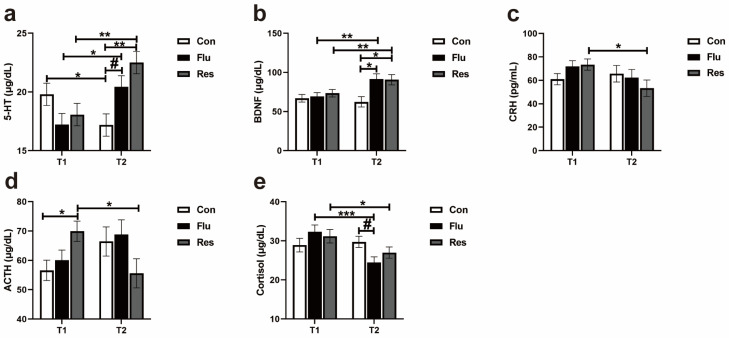
Effect of treatment administrations on 5-HT, BDNF, CRH, and HPA axis hormones at T1 and T2 in the dogs. (**a**) 5-HT, 5-hydroxytryptamine; (**b**) BDNF, brain derived neurotrophic factor; (**c**) CRH, corticotropin releasing hormone; (**d**) ACTH, adrenocorticotropic hormone; (**e**) cortisol. Data are presented as the mean ± SEM. The symbol (*) indicates a statistically significant difference between two groups (* *p* < 0.05, ** *p* < 0.01, and *** *p* < 0.001), and the symbol (#) represents a difference tendency (*p* < 0.10).

**Figure 3 antioxidants-14-00195-f003:**
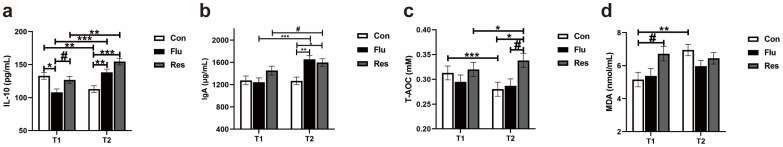
Effect of treatment administrations on the levels of anti-inflammatory cytokine, IgA, and antioxidant capacity at T1 and T2 in the dogs. (**a**) IL-10, interleukin 10; (**b**) IgA, immunoglobulin A; (**c**) T-AOC, total antioxidative capacity; (**d**) MDA, malondialdehyde. Data are presented as the mean ± SEM. The symbol (*) indicates a statistically significant difference between two groups (* *p* < 0.05, ** *p* < 0.01, and *** *p* < 0.001), and the symbol (#) represents a difference tendency (*p* < 0.10).

**Figure 4 antioxidants-14-00195-f004:**
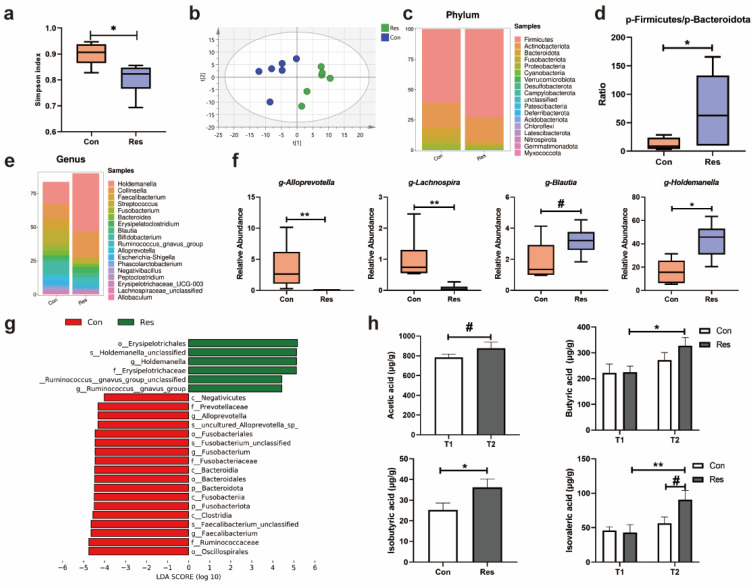
Effect of Res on gut microbiome at T2 and fecal SCFAs and BCFAs at T1 and T2 in the dogs. (**a**) Simpson index; (**b**) PCA (principal component analysis) plot of the microbiota; (**c**) histogram of the relative abundance at the phylum level; (**d**) ratio of the abundance of Firmicutes to Bacteroidota; (**e**) histogram of the relative abundance at the genus level; (**f**) relative abundances of *Alloprevotella*, *Lachnospira*, *Blautia*, and *Holdemanella*; (**g**) the LEfSe (Linear discriminant analysis Effect Size) analysis between the Con and Res groups; (**h**) SCFAs and BCFAs. The symbol (*) indicates a statistically significant difference between two groups (* *p* < 0.05 and ** *p* < 0.01), and the symbol (#) represents a difference tendency (*p* < 0.10).

**Figure 5 antioxidants-14-00195-f005:**
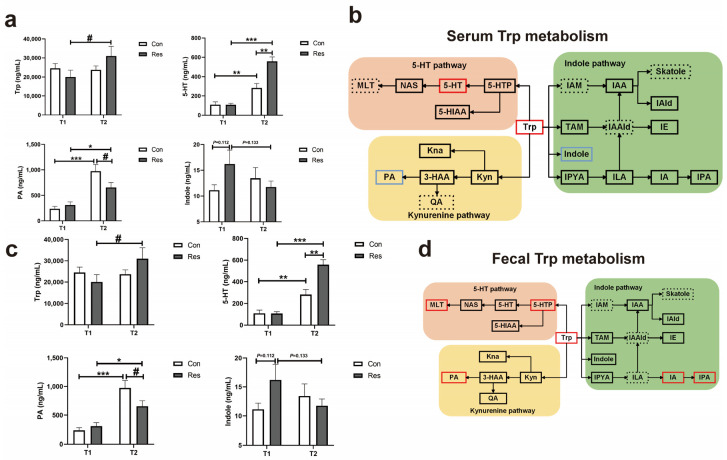
Effect of Res on the serum and fecal Trp metabolisms at T2 in the dogs. (**a**) Trp metabolism in the serum; (**b**) graph depicting the changes in the Trp metabolites in the serum; (**c**) Trp metabolism in the feces; (**d**) graph depicting the changes in the Trp metabolites in the feces. Black box: no statistical difference; red box: increase; blue box: decrease; dotted box: less than three sample were detected. 3-HAA, 3-hydroxyanthranilic acid; 5-HIAA, 5-hydroxyindoleacetic acid; 5-HT, 5-hydroxytryptamine; 5-HTP, 5-hydroxytryptophan; IA, indole acrylic acid; IAA, indole-3-acetic acid; IAAld, indole-3-acetaldehyde; IAld, indole-3-aldehyde; IAM, indole-3-acetamide; IE, indole ethanol; ILA, indole-3-lactic acid; Indole, indole; IPA, 3-indolepropionic acid; IPYA, indole-3-pyruvate; KNA, kynurenic acid; Kyn, kynurenine; MLT, melatonin; NAS, N-acetyl-5-hydroxytryptamine; PA, picolinic acid; QA, quinolinic acid; TAM, tryptamine; Trp, tryptophan. Data are presented as the mean ± SEM. The symbol (*) indicates statistically a significant difference between two groups (* *p* < 0.05, ** *p* < 0.01, and *** *p* < 0.001), and the symbol (#) represents a difference tendency (*p* < 0.10).

**Figure 6 antioxidants-14-00195-f006:**
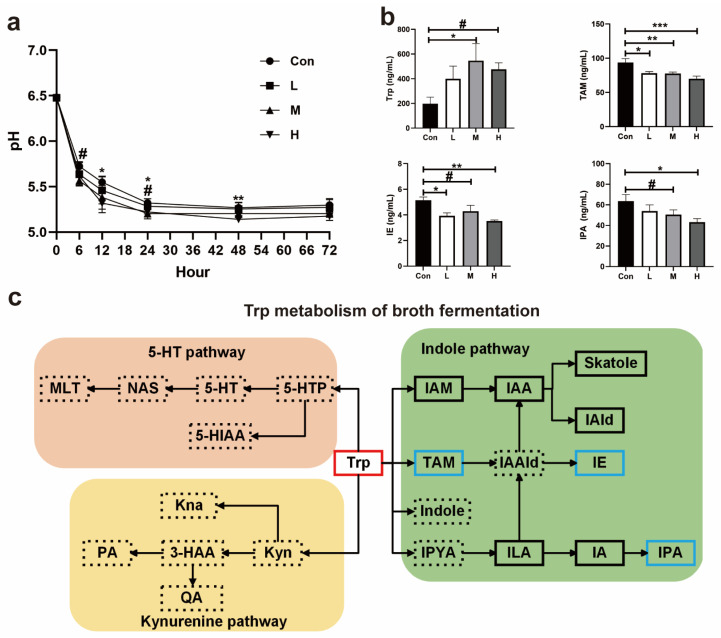
Effect of Res on the Trp metabolism in the broth fermentation. (**a**) pH of the fermentation broth; (**b**) Trp metabolism in the fermentation broth; (**c**) graph depicting the changes in the Trp metabolites in the fermentation broth. Black box: no statistical difference; red box: increase; blue box: decrease; dotted box: less than three samples were detected. 3-HAA, 3-hydroxyanthranilic acid; 5-HIAA, 5-hydroxyindoleacetic acid; 5-HT, 5-hydroxytryptamine; 5-HTP, 5-hydroxytryptophan; IA, indole acrylic acid; IAA, indole-3-acetic acid; IAAld, indole-3-acetaldehyde; IAld, indole-3-aldehyde; IAM, indole-3-acetamide; IE, indole ethanol; ILA, indole-3-lactic acid; Indole, indole; IPA, 3-indolepropionic acid; IPYA, indole-3-pyruvate; KNA, kynurenic acid; Kyn, kynurenine; MLT, melatonin; NAS, N-acetyl-5-hydroxytryptamine; PA, picolinic acid; QA, quinolinic acid; TAM, tryptamine; Trp, tryptophan. Data are presented as the mean ± SEM. The symbol (*) indicates a statistically significant difference between two groups (* *p* < 0.05, ** *p* < 0.01, and *** *p* < 0.001), and the symbol (#) represents a difference tendency (*p* < 0.10).

**Figure 7 antioxidants-14-00195-f007:**
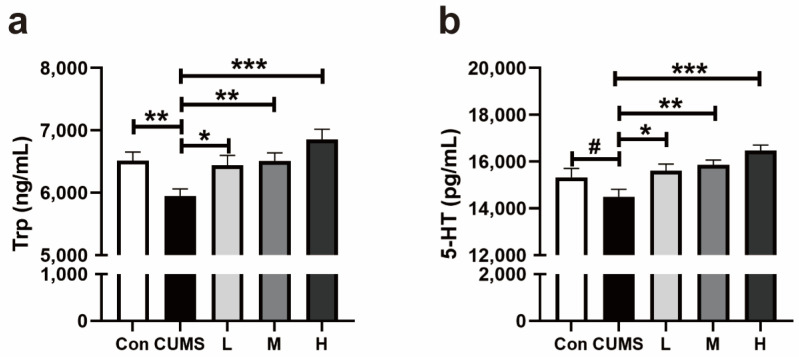
Effect of Res on the Trp and 5-HT in the whole brains in mice. (**a**) Trp, tryptophan; (**b**) 5-HT, 5-hydroxytryptamine. Data are presented as the mean ± SEM. The symbol (*) indicates a statistically significant difference between two groups (* *p* < 0.05, ** *p* < 0.01, and *** *p* < 0.001), and the symbol (#) represents a difference tendency (*p* < 0.10).

**Figure 8 antioxidants-14-00195-f008:**
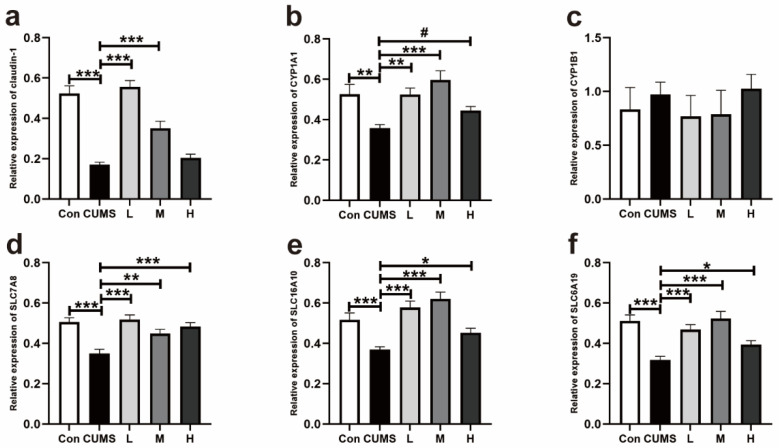
Effect of Res on the expression of genes related to the tight junction protein, aryl hydrocarbon receptor (AhR), and Trp transporters of the colons in mice. (**a**) *claudin-1*; (**b**) *CYP1A1*; (**c**) *CYP1B1*; (**d**) *SLC7A8*; (**e**) *SLC16A10*; (**f**) *SLC6A19*. Data are presented as the mean ± SEM. The symbol (*) indicates a statistically significant difference between two groups (* *p* < 0.05, ** *p* < 0.01, and *** *p* < 0.001), and the symbol (#) represents a difference tendency (*p* < 0.10).

## Data Availability

The 16S rRNA data were deposited in the NC-BI repository, accession number: https://www.ncbi.nlm.nih.gov/, accessed on 10 March 2023, PRJNA1035619. The metabolomic data were deposited in the EMBL-EBI MetaboLights repository with the identifiers MTBLS8902 and M-TBLS8941, with accession numbers corresponding to the following: https://www.ebi.ac.uk/metabolights/MTBLS8902 and https://www.ebi.ac.uk/metabolights/MTBLS8941, accessed on 10 March 2023.

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
