# Peer review of "Resveratrol Ameliorates Chronic Stress in Kennel Dogs and Mice by Regulating Gut Microbiome and Metabolome Related to Tryptophan Metabolism"

_antioxidants, 2025, doi:10.3390/antiox14020195_

Round 1
Reviewer 1 Report
Dear Authors,
It is evident that you have invested a great deal of effort into the experiment, analyses, and preparation of the article. However, in my opinion, the Introduction does not clearly explain why you decided to conduct Experiments 1 and 2 simultaneously. As a result, it is difficult to follow the structure and the possible connection between the results and the discussion.
- Please explain the concentrations of Res and Fluoxetine used in the study.
- Section 2.4.1: The ORF in dogs is described briefly, but not for mice. Please add this information. It is confusing because Table S2 is titled Dogs behaviors assessed in ORF, yet the first sentence of the Discussion states, OFT and FST are commonly used to assess stress and despair in rodents. As a result, it is difficult to follow what was done in mice versus dogs and how the results are connected. Adding clear labels in the figures to indicate whether the results pertain to dogs or mice would be helpful.
- Please explain the meaning of the stars in the figures.
- Table 1: Which nutritional recommendations did you follow? What about ME and the premix?
- Did you find any correlation between the results in serum and feces?
- Please check the legends in Figures S1b and S2d.
- All references have two similar numbers.
Author Response
Report
- Does the introduction provide a comprehensive yet concise overview about the state of knowledge in the area of research?
The authors have not sufficiently explained why they decided to conduct Experiments 1 and 2 simultaneously.
Response: The purpose of this experiment is to explore the mechanisms by which resveratrol alleviates chronic stress in dogs. In our experiments with dogs, we found that resveratrol has the potential to relieve stress, and this effect may be related to changes in tryptophan metabolism, especially the levels of tryptophan in the hindgut and blood. According to previous literatures, the levels of 5-HT in the brain are associated with the improvement of stress, and the levels of 5 - HT are determined by the concentration of tryptophan in the central nervous system. Additionally, tryptophan in the central nervous system is related to the levels of tryptophan in the hindgut. Therefore, we hypothesized that the improvement of stress in dogs is associated with the increased levels of 5-HT and tryptophan in the central nervous system. The alteration in tryptophan levels in the central nervous system might be attributed to the transport of tryptophan from the hindgut to the central nervous system via amino acid transporters in the intestinal tract. Considering the requirements of animal welfare and experimental costs, we are unable to collect gut and brain tissues from dogs to verify the above hypothesis. Therefore, we selected mice that are also under chronic stress as a model to collect gut and brain samples to detect the expression of tryptophan transporters and the levels of tryptophan and 5-HT in the brain to confirm whether the increase in 5-HT levels in the brain is related to the reduced consumption of tryptophan in the hindgut (line 53-59, line 64-67).
- Are the results presented clearly and in sufficient detail, are the conclusions supported by the results and are they put into context within the existing literature?
It is difficult to follow what was done in mice versus dogs and how the results are connected.
Response: Thank you for your comments. We have added some necessary sentences to make the results of Experiment 1 and Experiment 2 more closely related and easier to understand. (line 406-413).
- Is the quality and presentation of the figures satisfactory?
Adding clear labels in the figures to indicate whether the results pertain to dogs or mice would be helpful. Also, the meaning of the stars in the figures should be explained.
Response: Thank you for the reminder. We have added clear labels and explained the meaning of the stars in the figures.
Major comment
Dear Authors,
It is evident that you have invested a great deal of effort into the experiment, analyses, and preparation of the article. However, in my opinion, the Introduction does not clearly explain why you decided to conduct Experiments 1 and 2 simultaneously. As a result, it is difficult to follow the structure and the possible connection between the results and the discussion.
Response: The purpose of this experiment is to explore the mechanisms by which resveratrol alleviates chronic stress in dogs. In our experiments with dogs, we found that resveratrol has the potential to relieve stress, and this effect may be related to changes in tryptophan metabolism, especially the levels of tryptophan in the hindgut and blood. According to previous literatures, the levels of 5-HT in the brain are associated with the improvement of stress, and the levels of 5 - HT are determined by the concentration of tryptophan in the central nervous system. Additionally, tryptophan in the central nervous system is related to the levels of tryptophan in the hindgut. Therefore, we hypothesized that the improvement of stress in dogs is associated with the increased levels of 5-HT and tryptophan in the central nervous system. The alteration in tryptophan levels in the central nervous system might be attributed to the transport of tryptophan from the hindgut to the central nervous system via amino acid transporters in the intestinal tract. Considering the requirements of animal welfare and experimental costs, we are unable to collect gut and brain tissues from dogs to verify the above hypothesis. Therefore, we selected mice that are also under chronic stress as a model to collect gut and brain samples to detect the expression of tryptophan transporters and the levels of tryptophan and 5-HT in the brain to confirm whether the increase in 5-HT levels in the brain is related to the reduced consumption of tryptophan in the hindgut (line 53-59, line 64-67).
Detail comments
Comment 1. Please explain the concentrations of Res and Fluoxetine used in the study.
Response 1: As we stated in line 102-103, we referred to previous studies on stress in mice. We calculated the dosage based on the body surface area of mice and dogs to derive the dosage used.
Comment 2. Section 2.4.1: The ORF in dogs is described briefly, but not for mice. Please add this information. It is confusing because Table S2 is titled Dogs behaviors assessed in ORF, yet the first sentence of the Discussion states, OFT and FST are commonly used to assess stress and despair in rodents. As a result, it is difficult to follow what was done in mice versus dogs and how the results are connected. Adding clear labels in the figures to indicate whether the results pertain to dogs or mice would be helpful.
Response 2: We have added the information of OFT and FST (line 148-156). Because we conducted the open-field test in both dogs and mice to assess their stress levels, the experimental procedures were not identical. By evaluating the behavior of dogs in the open-field test, we discovered that resveratrol could alleviate stress in dogs. Subsequently, we utilized a chronically stressed mouse model to further elucidate the way by which tryptophan in the hindgut contributed to circulating tryptophan. At this stage, we performed the open-field test on mice to confirm that resveratrol also has a stress-relieving effect on mice. We have added clear labels.
Comment 3. Please explain the meaning of the stars in the figures.
Response 3: We have now explained the meaning of the stars in the figures.
Comment 4. Table 1: Which nutritional recommendations did you follow? What about ME and the premix?
Response 4: Based on information provided by the manufacturer, the basal diet met the nutrient recommendations by the Association of American Feed Control Officials (AAFCO, 2017) for dogs (line 94-95). We did not measure ME due to limitations of the experimental design and feeding environment. All three groups of dogs in Experiment 1 were fed the same basal diet, so ME should not have affected the results of the experiment. The premixes of the basal diets are labelled at the bottom of Table S1.
Comment 5. Did you find any correlation between the results in serum and feces?
Response 5: Based on the results of the metabolome in Experiment 1, we found that resveratrol increased the levels of tryptophan in the serum and feces of dogs (line 339-340). Through Experiment 2, we speculated that the increase in serum tryptophan levels might be related to the increase in tryptophan levels in the gut (line 440-444).
Comment 6. Please check the legends in Figures S1b and S2d.
Response 6: Two-way repeated measure analysis of variance with Bonferroni adjustment for multiple comparisons was performed to analyze the differences among groups (i.e., treatment administrations; TRT) at different time points (i.e., time) in this study. We found that only the main effect of time on the duration of escape attempt (Figure S1b) and the level of dopamine (Figure S2d) was significant. The effect of time × TRT was not significant. So, we only showed the values of the two indicators at two time points. We also added the symbol meaning (*) in the legends.
Comment 7. All references have two similar numbers.
Response 7: We have corrected the format of the references to avoid any errors.

Reviewer 2 Report
This study investigated the effects of a basal diet supplemented with Resveratrol (Res) for 35 days in 18 kennel dogs and in mice.
The objectives have to be improved and the authors have to address the main questions which should be answered through this study.
Limitations may be included as total RNAseq data would have been an option to look on expression levels of a larger number of genes.
Conclusions from the mice experiments are missing. In addition, which additional insights were gained through these experiments.
M& M:
please give the descriptions in section 2.4 separately by experiment 1 and 2.
Figure: please add the abbreviations used in this Figure to the legend. Add also T1 and T2 in Figure 1.
Author Response
Reviewer 2:
Report
- Does the title describe the article's topic with sufficient precision?
The study includes dogs and mice. The title does not reflect the mice experiments.
Response: We have added “and mice” after “kennel dogs” in title.
Does the introduction provide a comprehensive yet concise overview about the state of knowledge in the area of research?
Justification of experiments in dogs should br worked out in a better way as well as the mice experiments. Objectives of the study have to be improved.
Response: The purpose of this experiment is to explore the mechanisms by which resveratrol alleviates chronic stress in dogs. In our experiments with dogs, we found that resveratrol has the potential to relieve stress, and this effect may be related to changes in tryptophan metabolism, especially the levels of tryptophan in the hindgut and blood. According to previous literatures, the levels of 5-HT in the brain are associated with the improvement of stress, and the levels of 5 - HT are determined by the concentration of tryptophan in the central nervous system. Additionally, tryptophan in the central nervous system is related to the levels of tryptophan in the hindgut. Therefore, we hypothesized that the improvement of stress in dogs is associated with the increased levels of 5-HT and tryptophan in the central nervous system. The alteration in tryptophan levels in the central nervous system might be attributed to the transport of tryptophan from the hindgut to the central nervous system via amino acid transporters in the intestinal tract. Considering the requirements of animal welfare and experimental costs, we are unable to collect gut and brain tissues from dogs to verify the above hypothesis. Therefore, we selected mice that are also under chronic stress as a model to collect gut and brain samples to detect the expression of tryptophan transporters and the levels of tryptophan and 5-HT in the brain to confirm whether the increase in 5-HT levels in the brain is related to the reduced consumption of tryptophan in the hindgut (line 53-59, line 64-67). The objective of this experiment was to evaluate the stress-relieving effects of resveratrol in dogs and mice, and to elucidate the possible mechanisms using a multi-omics approach (line 67-70).
Major comment
- This study investigated the effects of a basal diet supplemented with Resveratrol (Res) for 35 days in 18 kennel dogs and in mice.
The objectives have to be improved and the authors have to address the main questions which should be answered through this study.
Response: The objective of this experiment was to evaluate the stress-relieving effects of resveratrol in dogs and mice, and to elucidate the possible mechanisms using a multi-omics approach (line 67-70). The following results might address the main question of this experiment. In the present study, we found that resveratrol had a stress-relieving effect on chronic stress in dogs by improving immunity and antioxidant capacity. By using microbiomics and metabolomics, we hypothesized that the tryptophan metabolism related to gut microbes might be the pathway through which resveratrol exerts its stress-relieving effect. Since it was not feasible to obtain canine gut and brain tissues, we used a mouse model of stress to further elucidate the relationship between hindgut and circulating tryptophan. We found that resveratrol could reduce the consumption of intestinal tryptophan by gut microbes, thereby allowing more tryptophan to be transported from the gut to the circulatory system and the CNS for 5-HT synthesis.
- Limitations may be included as total RNAseq data would have been an option to look on expression levels of a larger number of genes.
Response: Thank you for your valuable suggestion. We have added it in the section of limitation (line 650-651).
- Conclusions from the mice experiments are missing. In addition, which additional insights were gained through these experiments.
Response: We found that resveratrol could reduce the consumption of intestinal tryptophan by gut microbes, thereby allowing more tryptophan to be transported from the gut to the circulatory system and the CNS for 5-HT synthesis from the mice experiment. We mentioned in the conclusion that the stress-relieving effect of resveratrol is associated to tryptophan metabolism related to gut microbiota, which already includes the results of Experiments 1 and 2 to some extent.
Detail comments
M& M:
Comment 1. please give the descriptions in section 2.4 separately by experiment 1 and 2.
Response 1: In this section, we described experiments done on dogs and mice, respectively.
Comment 2. Figure: please add the abbreviations used in this Figure to the legend. Add also T1 and T2 in Figure 1.
Response 2: We have added the abbreviations and T1 and T2 in Figure 1 (line 106-110).

Reviewer 3 Report
My main concern is that your experiments were performed in dogs but also in mice. You should clarify your experimental design and explain why both species were used. In title, you should add “and mice” after “kennel dogs”. In Material and Methods, you should specify which analyses were performed in dogs and mice. You should provide them in different sections.
L39: Please add a reference
L43: “Gu et al. [3] proposed that the…” Please delete “[3]” form L45.
L76, 81: “feed” instead of “food”
L91: What is the meaning of references [7] and [8] here?
L94: Please refer to Fig. 1A and 1B separately.
L102: “basal diet”
L111: “assigned” instead of “divided”
L178-183: This part should be removed in the section of “Statistical analysis”
L276: Please delete “Only”
L279: What about acetic and isobutyric acids?
L315: “is presented”
L339: Is section 3.1.8. part of experiment 1 (with groups CON, L, M and H)?
L395: In CUMS?
L428: “Niu et al. [22] reported that…” Please delete “[22]” from L429.
L434: “could alleviate”
L453: “Huang et al. [26] also reported…” Please delete “[26]” from L454.
L516: “Fan et al. [36] reported that…” Please delete “[36]” from L518.
L566: “Gao et al. [43] have…” Please delete “[43]” from L568.
Author Response
Reviewer 3:
Report
- Does the title describe the article's topic with sufficient precision?
In title, you should add “and mice” after “kennel dogs”
Response: We have added “and mice” after “kennel dogs” in title.
Major comments
My main concern is that your experiments were performed in dogs but also in mice. You should clarify your experimental design and explain why both species were used. In title, you should add “and mice” after “kennel dogs”. In Material and Methods, you should specify which analyses were performed in dogs and mice. You should provide them in different sections.
Response: The purpose of this experiment is to explore the mechanisms by which resveratrol alleviates chronic stress in dogs. In our experiments with dogs, we found that resveratrol has the potential to relieve stress, and this effect may be related to changes in tryptophan metabolism, especially the levels of tryptophan in the hindgut and blood. According to previous literatures, the levels of 5-HT in the brain are associated with the improvement of stress, and the levels of 5 - HT are determined by the concentration of tryptophan in the central nervous system. Additionally, tryptophan in the central nervous system is related to the levels of tryptophan in the hindgut. Therefore, we hypothesized that the improvement of stress in dogs is associated with the increased levels of 5-HT and tryptophan in the central nervous system. The alteration in tryptophan levels in the central nervous system might be attributed to the transport of tryptophan from the hindgut to the central nervous system via amino acid transporters in the intestinal tract. Considering the requirements of animal welfare and experimental costs, we are unable to collect gut and brain tissues from dogs to verify the above hypothesis. Therefore, we selected mice that are also under chronic stress as a model to collect gut and brain samples to detect the expression of tryptophan transporters and the levels of tryptophan and 5-HT in the brain to confirm whether the increase in 5-HT levels in the brain is related to the reduced consumption of tryptophan in the hindgut (line 53-59, line 64-67). We have added “and mice” after “kennel dogs” in title. And we have described experiments done on dogs and mice in the section of Material and Methods, respectively.
Detail comments
Comment 1. L39: Please add a reference
Response 1: We have added some relevant reference in line 39.
Comment 2. L43: “Gu et al. [3] proposed that the…” Please delete “[3]” form L45.
Response 2: Thank you for your careful check. Based on previous articles published in this journal, the way we have cited the literature here seems appropriate, so we have not made changes.
Comment 3. L76, 81: “feed” instead of “food”
Response 3: We have corrected “food” to “feed” in line 85 and 90.
Comment 4. L91: What is the meaning of references [7] and [8] here?
Response 4: We referred to the dosages of resveratrol and fluoxetine administered in the references [7] and [8] to set the doses of resveratrol and fluoxetine received by the dogs in Experiment 1.
Comment 5. L94: Please refer to Fig. 1A and 1B separately.
Response 5: The timeline and sampling points for Experiments 1 and 2 are Figure 1a and b. To make figure1 easier to understand, we explain the meaning of the elements in the Figure 1 below the figure (line 106-110).
Comment 6. L102: “basal diet”
Response 6: We have corrected “base diet” to “basal diet” in line 118.
Comment 7. L111: “assigned” instead of “divided”
Response 7: We have corrected “divided” to “assigned” in line 127.
Comment 8. L178-183: This part should be removed in the section of “Statistical analysis”.
Response 8: This part described the methods used to analyze the data of gut microbiome. Based on previous articles published in this journal, it seemed appropriate to present this part here, so we have not made changes.
Comment 9. L276: Please delete “Only”
Response 9: We have deleted “Only” in line 305.
Comment 10. L279: What about acetic and isobutyric acids?
Response 10: The levels of acetic and isobutyric acid in the feces of dogs in both groups were elevated at T2 compared to T1 (P = 0.07 and P = 0.04, Figure 4h). The above has been added to the results (line 308-309).
Comment 11. L315: “is presented”
Response 11: We have corrected “was presented” to “is presented” in line 346.
Comment 12. L339: Is section 3.1.8. part of experiment 1 (with groups CON, L, M and H)?
Response 12: Yes, this part is one of the results belonging to Experiment 1.
Comment 13. L395: In CUMS?
Response 13: Yes. As described in section 2.3.2, the three groups L, M, and H received the same CUMS treatment. The difference is that these three groups received different doses of resveratrol.
Comment 14. L428: “Niu et al. [22] reported that…” Please delete “[22]” from L429.
Response 14: Thank you for your careful check. Based on previous articles published in this journal, the way we have cited the literature here seems appropriate, so we have not made changes.
Comment 15. L434: “could alleviate”
Response 15: We have corrected “could alleviated” to “could alleviate” in line 482.
Comment 16. L453: “Huang et al. [26] also reported…” Please delete “[26]” from L454.
Response 16: Thank you for your careful check. Based on previous articles published in this journal, the way we have cited the literature here seems appropriate, so we have not made changes.
Comment 17. L516: “Fan et al. [36] reported that…” Please delete “[36]” from L518.
Response 17: Based on previous articles published in this journal, the way we have cited the literature here seems appropriate, so we have not made changes.
Comment 18. L566: “Gao et al. [43] have…” Please delete “[43]” from L568.
Response 18: Based on previous articles published in this journal, the way we have cited the literature here seems appropriate, so we have not made changes.

Round 2
Reviewer 1 Report
Dear authors,
thank you for considering the comments and for carefully addressing all the questions. I agree with all the additions and improvements and have no further comments.
No additional comments.
Reviewer 3 Report
Authors made the necessary amendments and I suggest the acceptance of their article
Ok